# Analysis of the Risk of Oral Squamous Cell Carcinoma in Patients with and without Recurrent Aphthous Stomatitis: A Retrospective Evaluation of Real-World Data of about 150,000 Patients

**DOI:** 10.3390/cancers14236011

**Published:** 2022-12-06

**Authors:** Moritz Hertel, Senem Birinci, Max Heiland, Robert Preissner, Susanne Nahles, Andrea-Maria Schmidt-Westhausen, Saskia Preissner

**Affiliations:** 1Department of Oral and Maxillofacial Surgery, Charité—Universitätsmedizin Berlin, Corporate Member of Freie Universität Berlin, Humboldt-Universität zu Berlin, and Berlin Institute of Health, Augustenburger Platz 1, 13353 Berlin, Germany; 2Institute of Physiology and Science-IT, Charité—Universitätsmedizin Berlin, Corporate Member of Freie Universität Berlin, Humboldt-Universität zu Berlin, and Berlin Institute of Health, Philippstr. 12, 10115 Berlin, Germany; 3Department of Periodontology, Oral Medicine and Oral Surgery, Charité—Universitätsmedizin Berlin, Corporate Member of Freie Universität Berlin, Humboldt-Universität zu Berlin, and Berlin Institute of Health, Aßmannshauser Str. 4–6, 14197 Berlin, Germany

**Keywords:** recurrent aphthous stomatitis, chronic aphthae, oral squamous cell carcinoma, oral cancer

## Abstract

**Simple Summary:**

Recurrent aphthous stomatitis (RAS) is a common disease of the mouth. We wanted to find out if due to inflammation RAS may contribute to oral cancer. We analyzed data from the TriNetX database from patients with and without RAS and set oral squamous cell carcinoma (OSCC) as outcome. We found an association between RAS and the development of OSCC. These findings need to be interpreted cautiously, as RAS is not classified as a premalignant condition to date, and our applied methods have some limitations. We conclude that further clinical research is necessary and oral ulcers suspicious of OSCC should undergo biopsy as early as possible.

**Abstract:**

Background: Recurrent aphthous stomatitis (RAS) is found among the most frequent diseases of the oral cavity. It is characterized by repeated formation of painful ulcers. The question has risen if due to potential tumor-promoting inflammation and sustaining proliferative signaling RAS may contribute to oral cancer. Accordingly, the aim of the study was to assess if an association of RAS and the development oral squamous cell carcinoma (OSCC) could be found in a larger cohort. As recurrent aphthous stomatitis is not classified as an oral potentially malignant disorder, it was assumed that the risk of OSCC did not differ between patients with (cohort I) and without RAS (cohort II). Methods: Retrospective clinical data of patients diagnosed with and without RAS (International Classification of Diseases (ICD)-10 code K12) within the past 20 years and a body mass index of 19–30 kg/m^2^ were retrieved from the TriNetX database to gain initial cohort 0. Subjects suffering from RAS were assigned to cohort I, whereby cohort II was obtained from the remaining individuals, and by matching for age, gender, as well as (history of) nicotine and alcohol dependence. After defining the primary outcome as “OSCC” (ICD-10 codes C00-C14), a Kaplan–Meier analysis was performed, and risk and odds ratios were calculated. Results: Of a total of 24,550,479 individuals in cohort 0, 72,845 subjects were each assigned to cohort I (females: 44,031 (60.44%); males: 28,814 (39.56%); mean current age (±standard deviation) = 35.51 ± 23.55 years) and II (females: 44,032 (60.45%); males: 28,813 (39.55%); mean current age (±standard deviation) = 35.51 ± 23.56 years). Among the cohorts I and II, 470 and 135 patients were diagnosed with OSCC within five years. The according risk of developing oral cancer was 0.65% and 0.18%, whereby the risk difference of 0.47% was highly significant (*p* < 0.0001; Log-Rank test). The RR and OR were calculated as 3.48 (95% confidence interval (CI) lower: 2.88 and upper: 4.21) and 3.50 (95% CI lower: 2.89 and upper: 4.24). Conclusions: Among the patients suffering from RAS, a significantly augmented risk of developing OSCC was found. However, it has to be emphasized that the recent literature does not provide any confirmatory evidence that supports the retrieved results. Furthermore, the findings need to be interpreted cautiously due to specific limitations that come along with the applied methods. It should thus far only be concluded that further research is necessary to evaluate hypotheses that may be retrieved from the obtained results. Despite this controversy, oral ulcers suspicious of OSCC should undergo biopsy. Trial Registration: Due to the retrospective nature of the study, no registration was necessary.

## 1. Introduction

Oral squamous cell carcinoma (OSCC) is the most frequent malignant neoplasia of the head and neck region [1]. The risk of OSCC both within the oral cavity and the larynx is augmented by smoking and/or alcohol consumption, despite OSCCs located in the pharynx being associated with infection with human papillomavirus (HPV), especially HPV-16 and HPV-18 [2]. The underlying mechanisms of cancer development are relatively well understood based on knowledge about fundamental cellular key alterations and evidence of histopathological progression from epithelial atypia through increasing stages of dysplasia [2,3]. Accordingly, several entities providing an augmented risk of OSCC have been identified and classified as oral potential malignant disorders by the World Health Organization (WHO) [4,5].

Recurrent aphthous stomatitis (RAS; synonyms: recurrent aphthous/oral ulceration and canker sore) is characterized by the repeated formation of painful ulcers in the oral cavity. Depending on the specific clinical presentation, minor aphthae, major aphthae, and herpetiform aphthae are differentiated. RAS is found among the most frequent diseases of the oral mucosa. Females are more often affected than males, and the onset is typically found during the second or third decade [6]. Both the frequency of recurrence and the duration of disease show vast interindividual differences. To date, the etiology remains widely unknown. Due to its association with Behcet’s disease, stimulated T cells and monocytes are suspected to be involved in an underlying immunopathogenic process [7,8]. Despite the fact that aphthae heal spontaneously, symptomatic therapies are available, for example local administration of corticosteroids [9]. Even though it has to be emphasized that the recent literature does not provide any evidence for an augmented risk of oral cancer in patients suffering from recurrent aphthae [4], the question arises if RAS might potentially establish tumor-promoting inflammation and trigger sustaining proliferative signaling. Both mechanisms are known to be involved in neoplasia genesis [3]. Accordingly, the purpose of the present study was to analyze if the risk of developing oral cancer differed between patients with and without recurrent aphthae. In the recent literature, this question has not yet been specifically addressed. Due to the fact that RAS is not classified as an oral potentially malignant disorder (OPMD) [4], it was hypothesized that the risk of developing OSCC was not significantly higher in patients suffering from RAS compared to individuals without recurrent aphthae.

The TriNetX Global Health Research Network (TriNetX, Cambridge, MA, USA) was chosen to retrieve related data, as it provides access to a significant number of medical records from more than 120 healthcare organizations (HCOs) from 19 countries. TriNetX is a real-world database intending to enable HCOs, contract research institutes, and biopharmaceutical companies to access and exchange longitudinal clinical data and to provide state-of-the-art statistical analytics. By December 2021, TriNetX had collected electronic medical records from more than 250 million individuals. The network has previously been used to research medical topics of worldwide interest, including the coronavirus disease 2019 (COVID-19) pandemic [10,11].

## 2. Patients and Methods

### 2.1. Data Acquisition, Allocation, and Matching

Any data displayed on the TriNetX Platform in aggregate form, or any patient-level data provided in a data set generated by the TriNetX Platform, only contains de-identified data as per the de-identification standard defined in Section §164.514(a) of the HIPAA Privacy Rule. The TriNetX database was searched for both patients who were diagnosed with RAS (International Classification of Diseases (ICD)-10 code K12) and for subjects who did not suffer from RAS. As the purpose of the study was to test the expressed hypothesis on real-world data, patients of all ages were eligible. Due to the fact that the number of retrieved individuals was more than 40,000,000, which by far exceeds the analysis capacity of TriNetX, the eligibility period was limited to the previous 20 years from the access date. Furthermore, patients were defined to be eligible if their most recently determined body mass index (BMI) was 19–30 kg/m^2^. All patients diagnosed with ICD-10 code K12 were assigned to cohort I (subjects suffering from RAS). Cohort II (individuals without RAS) was subsequently obtained from the remaining patients within cohort 0, and by propensity score matching as shown in the modified CONSORT flow chart (Figure 1). Specifically, stratified and balanced sub-cohorts across current age at index and gender distributions, as well as (history of) nicotine and alcohol dependence, were retrieved from the initial cohorts to mitigate confounder bias via the propensity score [12,13,14]. In order to replicate randomized conditions as closely as possible, one-to-one matching was applied. Subsequently, the obtained cohorts I and II were tested for diagnosis with OSCC (ICD-10 codes C00-14), in the case of cohort I after having been diagnosed with RAS. The final cohorts were furthermore tested for human immunodeficiency virus (HIV) diagnoses (ICD-10 codes Z21 and B20–24).

### 2.2. Data Analysis

After defining the primary outcome as “OSCC”, a Kaplan–Meier analysis was performed, and risk ratios (RR), as well as odds ratios (OR), were calculated for the respective cohorts. The time window to record outcome events was five years from the respective index event (initial diagnosis of RAS among cohort I and visit to an HCO for any other reason referring to cohort II). Outcomes were recorded on a daily interval. A medical record covering the complete five-year follow-up was mandatory for inclusion. Statistical analysis was performed using the Log-Rank test, whereby *p* ≤ 0.05 was defined as a significance threshold.

## 3. Results

### 3.1. Assessment, Allocation, and Matching

The TriNetX network was accessed on Wednesday, 22 December 2021. A total eligible number of 88,760,357 patients from 65 healthcare organizations were initially available from the database. After limiting the eligibility period to the last 20 years and restricting the inclusion to the (most recently determined) body mass index (BMI) being 19–30 kg/m^2^, a total of 24,550,479 individuals remained available for allocation to cohort 0 (of which 197,454 subjects were diagnosed with OSCC within five years after the index event). Of those, 72,845 patients were diagnosed with RAS (ICD-10 code K12) and assigned to cohort I (females: 44,031 (60.44%); males: 28,814 (39.56%); mean current age at index (±standard deviation) = 35.51 ± 23.55 years). The same number of subjects was assigned by matching of both groups to cohort II, as explained above (females: 44,032 (60.45%); males: 28,813 (39.55%); mean current age at index (±standard deviation) = 35.51 ± 23.56 years). After the matching process, the groups did not differ significantly neither in gender distribution nor age, nor in the frequencies of (history of) nicotine and alcohol dependence (*p* = 1.0, 0.99, 0.97, and 0.93; Log-Rank test). The obtained propensity score was 0.98. Table 1 shows the patient characteristics of the cohorts before and after matching.

### 3.2. Diagnosis of OSCC

Among the cohorts I and II, 470 and 135 subjects were diagnosed with OSCC (ICD-10 codes C00–C14) within five years corresponding to risks of developing oral cancer of 0.65% and 0.18%, respectively. The calculated risk difference of 0.47% was statistically highly significant (*p* < 0.0001). The related RR and OR were 3.48 (95% confidence interval (CI) lower: 2.88 and upper: 4.21) and 3.50 (95% CI lower: 2.89 and upper: 4.24), as shown in Figure 2. The Kaplan–Meier diagram is displayed in Figure 3.

### 3.3. Human Immunodeficiency Virus (HIV)

The influence of various confounders on the results was checked. This included HIV infection, as it was shown to be associated with an augmented risk of AIDS (acquired immunodeficiency syndrome)-defining as well as non-AIDS-defining virus-related cancers [15,16]. A population-based study among subjects infected with HIV found a standardized incidence ratio (SIR) of 1.64 for HPV-related oral cavity and pharynx cancer (95% CI lower: 1.46 and upper: 1.84). Furthermore, elevated risks of several non-virus-related cancers including non-HPV-related oral cavity and pharynx cancer (SIR = 2.20 (95% CI lower: 1.98 and upper: 2.45)) and lip cancer (SIR = 2.35 (95% CI lower: 1.43 and upper: 3.62)) were reported [17]. It was found that the occurrence of HIV (ICD-10 codes Z21 and B20) was homogenously distributed when comparing the cohorts I and II.

## 4. Discussion

The present work aimed at assessing the risk of developing oral cancer among patients suffering from RAS compared to subjects without recurrent aphthae. This study was the first one to address this topic by retrospectively analyzing real-world data from multiple centers to investigate larger cohorts. As RAS is not classified as a premalignant condition, it was assumed that the risk of oral cancer did not differ between both cohorts. Different from the expressed hypothesis, it was found that the risk of OSCC was significantly higher in individuals with RAS in relation to subjects with no recurrent aphthae. The authors are well aware that the obtained results are in contrast with the recent literature in terms of RAS being not classified as an OPMD [4], and at least there is very limited evidence that supports the findings from the present work. A population-based frequency-matched case–control study from Taiwan found RAS in combination with dry eye syndrome to be a risk factor for oral cancer (HR = 3.41 (95% CI lower: 1.69 and upper: 6.86)), especially among females aged 50–69 years (HR = 5.56 (95% CI lower: 1.70 and upper: 18.25)) [18]. Sadraro et al. at least discussed the involvement of oxidative stress in both RAS and OSCC, which possibly links both entities regarding their pathogenesis [19]. Furthermore, there is evidence indicating that RAS could be correlated with genetic alterations [20]. Accordingly, a study from Lu et al. revealed an altered expression of IncRNA Cancer Susceptibility Gene 2 (CASC2) in patients with RAS [21]. CASC2 is known to play an oncogenic role in OSCC as well as in other malignant tumors [22,23]. Together with the results from the present study, a new hypothesis may be formulated in terms of carefully assuming RAS to be a risk factor for OSCC.

However, the results cannot be interpreted critically enough, and conclusions should be made with the greatest caution. Along with the retrospective nature of the study come specific limitations which need to be addressed. First of all, neoplastic ulcers can mimic benign ulcerative lesions, including aphthae [24]. Theoretically, there is a potential risk that OSCCs or precursor lesions were wrongly diagnosed as RAS, and the initial diagnoses were not revised after the (delayed) diagnosis of OSCC.

Furthermore, RAS is diagnosed based on the clinical presentation of aphthae and a positive anamnesis for their repeated recurrence. However, other entities can mimic RAS or at least present as aphthous lesions. Again, wrong diagnoses can be the result. As an example, secondary syphilis is known to potentially manifest as aphthous enanthema [25]. Even though syphilis is relatively rare in industrialized countries today, it was frequently described as a re-emerging global public health problem, particularly among men having sex with men [26,27]. Nonetheless, it is actually not classified as an OPMD [4]. Oral lichen planus (OLP), which is also classified as a premalignant condition, can come along with erosions and ulcers as well [4]. Even though those entities can be distinguished from RAS via specific diagnostic measures (serological testing in the case of syphilis and based on histopathological examination after biopsy taking in the case of OLP), it cannot be retrospectively assessed if the relevant differential diagnoses were safely excluded. For that reason, it cannot be safely excluded that cases of RAS found in the database were in fact other premalignant entities causing an augmented incidence of OSCCs among the patients in cohort I.

Another potential point of criticism arises from the very limited availability of details on the smoking behavior and alcohol consumption of the included patients. Thus far, TriNetX allows for screening of subjects with nicotine and alcohol dependence or a history of such. However, the respective terms might not have been consistently defined, leading to inhomogeneities in the retrieved data. Furthermore, neither the number of cigarettes smoked per day nor the pack-years were available for smokers. From the individuals classified as alcohol dependent, it was not reported how many units of alcohol were consumed, or since which period of time. All these features do have a well investigated impact on the risk of oral cancer [28]. Accordingly, the applied approach does imply a certain risk of bias. However, it can be carefully assumed that matching for age, gender, and (history of) smoking tobacco and drinking alcohol might have leveled out distribution differences in risk factors, at least to a limited extent. Regarding data on HIV, the applied retrospective approach implies analogous limitations. Assumedly, testing for HIV has not been carried out routinely. As a result, a higher number of patients with undetected HIV infections might eventually have been allocated to cohort I compared to cohort II. This might at least theoretically have contributed to the obtained results. No sufficient data on HPV status were available due to inhomogeneities regarding the methods applied to diagnose HPV. Despite the fact that this is a limitation of the study, as HPV plays a role in the development of oral cancer, it needs to be emphasized that HPV is not associated with RAS. The use of steroids among cohort I was not consistently documented, as a proportion of the patients underwent symptomatic treatment of RAS at their dentists, and therefore outside of the HCO. Furthermore, no geographical data were available, whereby the vast majority of the HCOs within the TriNetX network are located in the United States of America and in China. Assumedly, the majority of the subjects involved origin from the respective countries. Nevertheless, inhomogeneous healthcare standards and/or patient behavior cannot be safely excluded.

With the utmost caution, it can be discussed that RAS may provide proinflammatory signaling, which is known to be a potential trigger of cancer cell formation [3]. Based on the consecutive concept of cellular key alterations leading to epithelial atypia followed by dysplasia and tumor development, there is at least a theoretical possibility that recurrent aphthae might contribute to oral cancer. Basically, this was the idea behind the present work, even though the retrieved results were not expected. However, certain aspects which were in the focus of research regarding the etiology of RAS should be discussed concerning the presented findings. Again, the authors highlight that the retrieved results do not allow for specific conclusions, but only for careful assumptions to be made. Recurrent aphthae were partially found to be associated with deficiency syndromes, specifically celiac disease and B12 deficiency [29]. In the recent literature limited evidence is available, indicating an augmented risk of gastrointestinal cancer related to celiac disease [30,31]. Nevertheless, a correlation between B12 deficiency and cancer remains a matter of strong controversy [32]. Despite deficiency diseases, an association of RAS with periodic fever, aphthous stomatitis, pharyngitis, and adenitis syndrome (PFAPA) was discussed [29]. However, no evidence for an augmented risk of cancer was found for PFAPA in the recent literature. Other studies on the etiology of recurrent aphthae investigated a potential genetic background. In particular, the interleukin (IL)-1 beta polymorphism was described to be potentially related to RAS [33]. Accordingly, an association of different polymorphisms with different forms of cancer has been discussed [34,35]. Nevertheless, recent studies indicate a correlation of the IL-4 gene but not the IL-1 beta gene polymorphism with oral cancer [36].

Despite the problems with putting the findings into the context of the available literature, there is one conclusion which appears to be legitimate. Delayed diagnosis of OSCC or its precursors leads to the necessity of more aggressive therapies and decreases the survival rate of the affected patients [37,38]. Hence, ulcers in the oral cavity should undergo biopsy, at least in suspicious cases, which means persistence despite 10 to 14 days of treatment, absence of pain, and the presence of specific risk factors for OSCC. This approach appears to be appropriate for the management of oral ulcerous lesions including RAS.

In order to overcome the aforementioned limitations, future research might consider using a prospective approach applying standardized diagnostic measures, to evaluate if the presented results can be confirmed thus far.

## Figures and Tables

**Figure 1 cancers-14-06011-f001:**
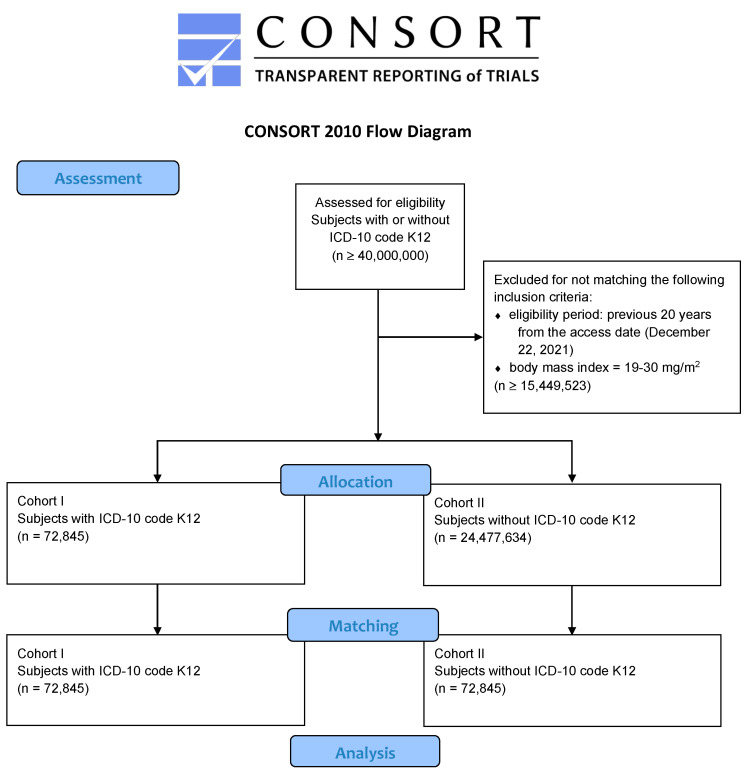
Modified CONSORT flow chart. Note: the number of eligible subjects was ≥40,000,000, which is above the analysis capacity of TriNetX. As a consequence, additional inclusion criteria were necessary to restrict the enclosure of patients.

**Figure 2 cancers-14-06011-f002:**
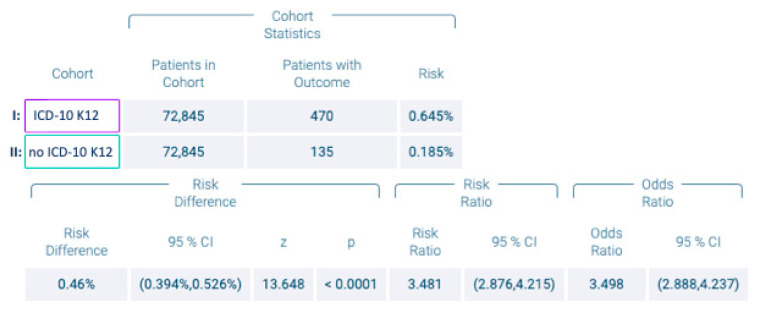
Number of patients with and without recurrent aphthous stomatitis (RAS; ICD-10 code K12), and risk of oral squamous cell carcinoma.

**Figure 3 cancers-14-06011-f003:**
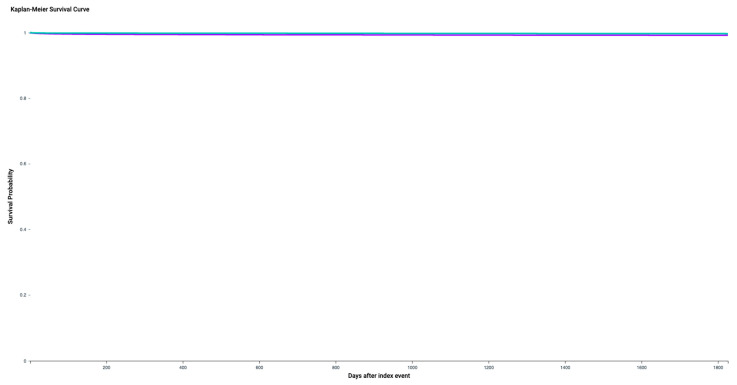
Kaplan–Meier diagram of the cohorts I (patients with recurrent aphthous stomatitis (RAS; ICD-10 code K12) and II (subjects without RAS). cohort I = purple; cohort II = green.

**Table 1 cancers-14-06011-t001:** Patient characteristics before and after matching of cohorts I (ICD-10 code K12) and II (without ICD-10 code K12).

Patients (*n*)	Before Matching				After Matching			
	Cohort I	Cohort II	*p*-Value	Standardized Mean Difference	Cohort I	Cohort II	*p*-Value	Standardized Mean Difference
Total	72,845	24,477,634			72,845	72,845		
Males	28,814 (39.56%)	11,213,208 (46.0%)			28,814 (39.66%)	28,813 (39.55%)		
Females	44,031 (60.44%)	13,264,426 (54.0%)	<0.0001	0.1267	44,031 (60.44%)	44,032 (60.45%)	1.0	0.0
Mean age at Index	35.51	38.15	<0.0001	0.1109	35.51	35.51	0.99	0.0
Standard deviation	23.55	24.02			23.55	23.56		
Minimum	0	0			0	0		
Maximum	90	90			90	90		
Nicotine abuse	4089 (5.61%)	189,264 (0.77%)	<0.0001	0.2779306	4089 (5.61%)	4092 (5.61%)	0.97	0.0002
Alcohol abuse	578 (0.79%)	24,620 (0.10%)	<0.0001	0.10400509	578 (0.79%)	575 (0.79%)	0.93	0.0005

Abbreviations: ICD, International Classification of Diseases. Note: Percentage refers to gender distribution within the respective cohorts. *p*-value refers to comparison between both cohorts (Log-Rank test).

## Data Availability

The data sets used and analyzed can be retrieved from the TriNetX network (https://trinetx.com), access date: 19 December 2021. If no access is available, the data sets can be retrieved from the corresponding author based on reasonable request.

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
