# Peer review of "Analysis of the Risk of Oral Squamous Cell Carcinoma in Patients with and without Recurrent Aphthous Stomatitis: A Retrospective Evaluation of Real-World Data of about 150,000 Patients"

_cancers, 2022, doi:10.3390/cancers14236011_

Round 1
Reviewer 1 Report
The article is original, interesting and well written , but the retrospective nature and the biases that are highlighted in the RAS group make in my opinion this study of not high impact for such a prestigious journal as Cancers
Author Response
Reviewer #1
The article is original, interesting and well written, but the retrospective nature and the biases that are highlighted in the RAS group make in my opinion this study of not high impact for such a prestigious journal as Cancers
Our response: We kindly thank the reviewer for this comment. We strongly believe that the obtained results are of interest for the readership of Cancers. The strength of studies involving real-world data of large cohorts is testing of hypotheses, especially in fields where only limited data is available. This was the specific purpose of our study. We furthermore think that our findings will encourage further research (see: comment of Reviewer #2). Therefore, we kindly ask the reviewer to rethink his or her decision.
Revised text: None.
Reviewer #2
The article is well written and the focus of the article is right. In my opinion your study was performed correctly and it talks about a condition, recurrent aphthous stomatitis, that is pretty common in the daily practice. That’s why I believe it could give some new useful data, also because there is a lack of recent information regarding the main topic of this manuscript in the current literature. It could be a good start for future researches. English language and style are also fine. My only suggestion is to explain more properly why the recent literature is in contrast with the authors (line 165) and, if possible, to add more references about it. The submitted manuscript is clear and complete so I think it could be published with a minor revision.
Our response: We kindly thank the reviewer for this comment. According to the reviewers’ kind suggestion we added a possible explanation on why our findings are in contrast with the recent literature. We furthermore added the requested literature.
Revised text: Discussion, References.
Reviewer 2 Report
Dear Authors,The article is well written and the focus of the article is right. In my opinion your study was performed correctly and it talks about a condition, recurrent aphthous stomatitis, that is pretty common in the daily practice. That’s why i believe it could give some new useful data, also because there is a lack of recent information regarding the main topic of this manuscript in the current literature. It could be a good start for future researches. English language and style are also fine. My only suggestion is to explain more properly why the recent literature is in contrast with the authors (line 165) and, if possible, to add more references about it. The submitted manuscript is clear and complete so i think it could be published with a minor revision.
Author Response
Reviewer #2
The article is well written and the focus of the article is right. In my opinion your study was performed correctly and it talks about a condition, recurrent aphthous stomatitis, that is pretty common in the daily practice. That’s why I believe it could give some new useful data, also because there is a lack of recent information regarding the main topic of this manuscript in the current literature. It could be a good start for future researches. English language and style are also fine. My only suggestion is to explain more properly why the recent literature is in contrast with the authors (line 165) and, if possible, to add more references about it. The submitted manuscript is clear and complete so I think it could be published with a minor revision.
Our response: We kindly thank the reviewer for this comment. According to the reviewers’ kind suggestion we added a possible explanation on why our findings are in contrast with the recent literature. We furthermore added the requested literature.
Revised text: Discussion, References.
Reviewer 3 Report
This is a very interesting study. Applying AI is novel to identify additional risk factors for oral carcinogenesis.
Few minor comments.
1. In the abstract rather than saying "Hypothesis was not confirmed" it is best to give the direct result. ie a statistically significant association was found....
2, It is not clear how much follow-up time was allowed for the two cohorts to determine the development of OSCC or not.
3. Was the steroid use in treating RAU documented in records? Kindly make a comment.
4, reference to the WHO at the end of the 1st paragraph in the introduction is not correct. These two cited papers are not originating from the WHO. Suggest use of the WHO Collaborating Centre report (2007) in J Oral Pathology and Medicine 2007; Vol 36; 575-80.
5. The term "Potentially Malignant Disorders" is preferred to "Precancerous lesions and conditions"
6. In 2.2. Data analysis; the authors refer to Kaplan-Meyer Analysis was performed. But K-M graphs are not supplied. Please include the K-M Graphs as figures in the publication.
Author Response
Reviewer #3
This is a very interesting study. Applying AI is novel to identify additional risk factors for oral carcinogenesis. Few minor comments.
- In the abstract rather than saying "Hypothesis was not confirmed" it is best to give the direct result. ie a statistically significant association was found....
Our response: We kindly thank the reviewer for this helpful suggestion. We changed the Abstract, as requested.
Revised text: Abstract (Conclusions).
2, It is not clear how much follow-up time was allowed for the two cohorts to determine the development of OSCC or not.
Our response: We thank the reviewer for this important question. The follow-up time (= time-window) was five years and only medical records covering the complete five-year period were eligible. We added this important information to the manuscript.
Revised text: Abstract (Results), Patients and Methods (Data Analysis), Results (Diagnosis of OSCC).
- Was the steroid use in treating RAU documented in records? Kindly make a comment.
Our response: This question is very interesting. Unfortunately, no sufficient data on the use of steroids was available as a proportion of the patients underwent symptomatic treatment of RAS outside of the HCO. This information was added to the manuscript.
Revised text: Discussion.
- Reference to the WHO at the end of the 1st paragraph in the introduction is not correct. These two cited papers are not originating from the WHO. Suggest use of the WHO Collaborating Centre report (2007) in J Oral Pathology and Medicine 2007; Vol 36; 575-80.
Our response: According to the reviewer’s kind suggestion we replaced the references by “Warnakulasuriya S, Kujan O, Aguirre-Urizar JM, et al. Oral potentially malignant disorders: A consensus report from an international seminar on nomenclature and classification, convened by the WHO Collaborating Centre for Oral Cancer. Oral Dis. Nov 2021;27(8):1862-1880. doi:10.1111/odi.13704” and “Warnakulasuriya S, Johnson NW, van der Waal I. Nomenclature and classification of potentially malignant disorders of the oral mucosa. J Oral Pathol Med. Nov 2007;36(10):575-80. doi:10.1111/j.1600-0714.2007.00582”.
Revised text: Introduction, References.
- The term "Potentially Malignant Disorders" is preferred to "Precancerous lesions and conditions"
Our response: We totally agree. Accordingly, the term “precancerous lesions and conditions” was changed to “oral potentially malignant disorders (OPMD)” in all parts of the manuscript.
Revised text: Abstract (Background), Introduction, Discussion.
- In 2.2. Data analysis; the authors refer to Kaplan-Meyer Analysis was performed. But K-M graphs are not supplied. Please include the K-M Graphs as figures in the publication.
Our response: As requested by the reviewer, we added the Kaplan-Meier diagram to the manuscript as “Figure 3”.
Revised text: Results (Diagnosis of OSCC).
Reviewer 4 Report
In their manuscript entitled “Analysis of the risk of oral squamous cell carcinoma in patients with and without recurrent aphthous stomatitis: a retrospective evaluation of real-world data of about 150,000 patients”, Hertel and colleagues analyze the correlation between recurrent aphthous stomatitis (RAS) and the later diagnosis of an OSCC in patients of the TriNetX database. The authors observed a significant correlation between a RAS diagnosis and an increased risk for an OSCC diagnosis. The manuscript is written understandable and comprehensible. However, I have major concerns on the methodology and the results, therefore would not recommend publication of the manuscript in the present form.
Here are my concerns in detail:
1) I have some troubles with the statistics used in this manuscript. I understand, that cohort I is defined as the RAS population, and that an age- and gender-matched subgroup was drawn from the overall Non-RAS population as cohort II. However, I do not understand why this procedure was only performed once, but not in a bootstrapping approach, as it would be easily possible with the vast Non-RAS cohort to perform appropriate series of resampling. I would recommend to re-think the statistics, and maybe – if not already undertaken – consult another statistician.
2) Also, it is possible to pre-select cohort II according to the follow-up time, which should be comparable to the one of cohort I. As it is stated, RAS is a syndrome mainly arising in patients 20 – 30 years old, while OSCC has its peak in the age of 50 – 60 years. How can it be avoided, that patients of cohort II have left in higher rates the HCO within the TriNetX database than patients in cohort I?
3) Furthermore, in Table 1, the age minima in cohort I and II are given as 0 years. Does it make sense to include newborns in this analysis? I would suggest to apply a lower age border of for instance 40 years.
4) Looking in the TriNetX database, I learned that most of the patients included were from the US (~22 mio), and the only other significant cohort was from China (~7 mio). Is it possible to subclassify according to geographical region to avoid different healthcare standards and/or different patient behavior? Maybe a US and a Chinese cohort? Is it possible to analyze where the RAS cases predominantly originated from?
5) What was the overall occurrence of OSCC in the TriNetX database cohort, and in the Non-RAS cohort? Did the authors perform a power analysis on the subjects to include?
6) I do not understand, why in subchapter 3.3 HIV was analyzed as confounder, as – to the best of my knowledge – an HIV infection is not associated with RAS nor OSCC. Why not check for human papilloma virus (HPV) – ICD-10 B97.7 or similar?
7) Given the assumption that the authors thoroughly revise the analyses, the discussion of course has to be adapted to potentially other results. Also, the authors should consider to add some newer literature on oral lesions/OSCC and syphilis, and comment on how severe this problem is really nowadays.
Minor comments:
8) Table 1: Nicotine/alcohol dependence à nicotine/alcohol abuse?
9) Appendix A – HCO = health care organization
Finally, I want to thank the authors for presenting this interesting approach in analyzing real-world health data for unconventional research questions. However, I stay with my criticism, and have the strong feeling that the manuscript and the analyses should be seen by at least one other statistician to improve the quality of the analyses.
Best regards.
Author Response
Reviewer #4
In their manuscript entitled “Analysis of the risk of oral squamous cell carcinoma in patients with and without recurrent aphthous stomatitis: a retrospective evaluation of real-world data of about 150,000 patients”, Hertel and colleagues analyze the correlation between recurrent aphthous stomatitis (RAS) and the later diagnosis of an OSCC in patients of the TriNetX database. The authors observed a significant correlation between a RAS diagnosis and an increased risk for an OSCC diagnosis. The manuscript is written understandable and comprehensible. However, I have major concerns on the methodology and the results, therefore would not recommend publication of the manuscript in the present form.
Here are my concerns in detail:
1) I have some troubles with the statistics used in this manuscript. I understand, that cohort I is defined as the RAS population, and that an age- and gender-matched subgroup was drawn from the overall Non-RAS population as cohort II. However, I do not understand why this procedure was only performed once, but not in a bootstrapping approach, as it would be easily possible with the vast Non-RAS cohort to perform appropriate series of resampling. I would recommend to re-think the statistics, and maybe – if not already undertaken – consult another statistician.
Our response: We thank the reviewer for this valuable remark. We chose propensity score matching over bootstrapping for the following reasons:
- both methods are of equal value, but propensity score matching is easier to conduct*
- propensity score matching is well-established for analyzing real-world data, and it is propagated as state-of-the-art method by TriNetX
However, we consulted another statistician to be sure that our approach was technically correct, as kindly suggested. The statistician confirmed that the method was correctly selected and applied, and provided the following supportive literature, which we also added to the manuscript:
* Geldof T, Popovic D, Van Damme N, Huys I, Van Dyck W. Nearest Neighbour Propensity Score Matching and Bootstrapping for Estimating Binary Patient Response in Oncology: A Monte Carlo Simulation. Sci Rep. Jan 22 2020;10(1):964. doi:10.1038/s41598-020-57799-w
* Desai RJ, Wyss R, Abdia Y, et al. Evaluating the use of bootstrapping in cohort studies conducted with 1:1 propensity score matching-A plasmode simulation study. Pharmacoepidemiol Drug Saf. Jun 2019;28(6):879-886. doi:10.1002/pds.4784
* Austin PC, Small DS. The use of bootstrapping when using propensity-score matching without replacement: a simulation study. Stat Med. Oct 30 2014;33(24):4306-19. doi:10.1002/sim.6276
Revised text: Patients and Methods (Data Acquisition, Allocation and Matching), References.
2) Also, it is possible to pre-select cohort II according to the follow-up time, which should be comparable to the one of cohort I. As it is stated, RAS is a syndrome mainly arising in patients 20 – 30 years old, while OSCC has its peak in the age of 50 – 60 years. How can it be avoided, that patients of cohort II have left in higher rates the HCO within the TriNetX database than patients in cohort I?
Our response: We thank the reviewer for this very important question. The time-window to record outcomes was five years, and only medical records covering the complete five-year period were eligible for inclusion. This important information was added to the relevant sections of the manuscript.
Revised text: Abstract (Results), Patients and Methods (Data Analysis), Results (Diagnosis of OSCC).
3) Furthermore, in Table 1, the age minima in cohort I and II are given as 0 years. Does it make sense to include newborns in this analysis? I would suggest to apply a lower age border of for instance 40 years.
Our response: We thank the reviewer for this kind suggestion. RAS and OSCC can both occur in young subjects, even if cases are rare. Furthermore, the given age was the age at index (in terms of the point in time in which a patient enters the five-year follow-up / time-window), not at outcome. As our purpose was to test the expressed hypothesis on “real-world data”, we decided not to limit our analysis to a certain age stratum. However, we took the reviewer’s helpful remark as an occasion to explain our rationale to include all ages. We furthermore clarified that the presented age was specifically the age at index.
Revised text: Patients and Methods (Data Acquisition, Allocation and Matching), Results (Assessment, Allocation and Matching), Table 1.
4) Looking in the TriNetX database, I learned that most of the patients included were from the US (~22 mio), and the only other significant cohort was from China (~7 mio). Is it possible to subclassify according to geographical region to avoid different healthcare standards and/or different patient behavior? Maybe a US and a Chinese cohort? Is it possible to analyze where the RAS cases predominantly originated from?
Our response: We thank the reviewer for this interesting question. Unfortunately, no geographical data was available. As a consequence, we added a respective statement to the manuscript and discussed that as a limitation of the study.
Revised text: Discussion.
5) What was the overall occurrence of OSCC in the TriNetX database cohort, and in the Non-RAS cohort? Did the authors perform a power analysis on the subjects to include?
Our response: At the access date TriNetX accounted for 197,454 patients with OSCC who met the inclusion criteria. This information was added to the manuscript. Accordingly, the non-RAS cohort accounted for 196,984 eligible cases of OSCC before the matching process. We agree that a power analysis would have been necessary to determine the size of the cohorts if only a certain proportion of the available cases within the RAS cohort would have been included (e.g. by limiting the analysis to a certain age stratum), but not as we proceeded. We furthermore suppose that the large number of included patients and the retrieved small confidence intervals substantiate the significance of the obtained results.
Revised text: Results (Assessment, Allocation and Matching).
6) I do not understand, why in subchapter 3.3 HIV was analyzed as confounder, as – to the best of my knowledge – an HIV infection is not associated with RAS nor OSCC. Why not check for human papilloma virus (HPV) – ICD-10 B97.7 or similar?
Our response: We thank the reviewer for his/her valuable suggestion. The rationale to check the frequency of HIV infection was that HIV was shown to be associated with an increased risk of several cancers for numerous reasons (co-infections with oncogenic viruses, such as human papillomavirus (HPV) or hepatitis viruses, and high prevalence of behaviors, such as smoking and alcohol intake). We added this explanation to the manuscript including a respective reference. Despite that we totally agree that checking for HPV status would have been favorable, as well. However, the available data were inhomogeneous regarding the applied methods to diagnose HPV, which is why we decided not to analyze HPV. In turn, we added a sentence to the discussion stating the lack of sound data regarding HPV.
Revised text: Results (Human Immunodeficiency Virus (HIV)), Discussion, References.
7) Given the assumption that the authors thoroughly revise the analyses, the discussion of course has to be adapted to potentially other results. Also, the authors should consider to add some newer literature on oral lesions/OSCC and syphilis, and comment on how severe this problem is really nowadays.
Our response: According to the reviewer’s valuable advice we revised all relevant parts of the manuscript and explained the rationale for the applied methods. We hope that our explanations are plausible and that he/she might rethink his/her rejection. Furthermore, we added some newer literature on OPMDs and further statements dealing with syphilis, as kindly suggested.
Revised text: Discussion, References.
Minor comments:
8) Table 1: Nicotine/alcohol dependence à nicotine/alcohol abuse?
Our response: We thank the reviewer for this kind advice. We changed the terms, accordingly.
Revised text: Table 1.
9) Appendix A – HCO = health care organization
Our response: We thank the reviewer for this kind advice. We corrected the spelling mistake.
Revised text: Appendix A.
Finally, I want to thank the authors for presenting this interesting approach in analyzing real-world health data for unconventional research questions. However, I stay with my criticism, and have the strong feeling that the manuscript and the analyses should be seen by at least one other statistician to improve the quality of the analyses.
Our response: We totally appreciate the constructive analysis of our article and thank the reviewer for his/her very helpful suggestions. As stated above, we consulted a second statistician and explained the rationale for the applied methods. We hope that the reviewer might rethink his/her decision.
Revised text: see above.
Round 2
Reviewer 4 Report
In their manuscript entitled “Analysis of the risk of oral squamous cell carcinoma in patients with and without recurrent aphthous stomatitis: a retrospective evaluation of real-world data of about 150,000 patients”, Hertel and colleagues analyze the correlation between recurrent aphthous stomatitis (RAS) and the later diagnosis of an OSCC in patients of the TriNetX database. The authors observed a significant correlation between a RAS diagnosis and an increased risk for an OSCC diagnosis. This is the first revision of the manuscript.
The authors have responded detailed and comprehensible to all my comment. Due to the revisions applied, I have a better understanding of the methodology and rationale of the authors. Therefore, I re-considered my previous - rather harsh - recommendation for this manuscript, and recommend acceptance for publication.
Minor comment:
Regarding #6: The authors should be more specific in l.172, stating that Hernandez-Ramirez et al., 2017 observed a significant increased standardized incidence ratio in HIV-infected persons for non-HPV oral cavity cancer. (see table 2 in the respective article).
Finally, I want to thank the authors for presenting this interesting approach in analyzing real-world health data for unconventional research questions. Best regards.
Author Response
In their manuscript entitled “Analysis of the risk of oral squamous cell carcinoma in patients with and without recurrent aphthous stomatitis: a retrospective evaluation of real-world data of about 150,000 patients”, Hertel and colleagues analyze the correlation between recurrent aphthous stomatitis (RAS) and the later diagnosis of an OSCC in patients of the TriNetX database. The authors observed a significant correlation between a RAS diagnosis and an increased risk for an OSCC diagnosis. This is the first revision of the manuscript.
The authors have responded detailed and comprehensible to all my comment. Due to the revisions applied, I have a better understanding of the methodology and rationale of the authors. Therefore, I re-considered my previous - rather harsh - recommendation for this manuscript, and recommend acceptance for publication.
Minor comment:
Regarding #6: The authors should be more specific in l.172, stating that Hernandez-Ramirez et al., 2017 observed a significant increased standardized incidence ratio in HIV-infected persons for non-HPV oral cavity cancer. (see table 2 in the respective article).
Our response: We kindly thank the reviewer for having re-considered his/her decision after having read our response to his/her review. We furthermore think that our manuscript benefited much from the expressed criticism. According to the reviewers request we specified the risk of oral cancer in HIV infected subjects and added two more references.
Revised text: Results (Human Immunodeficiency Syndrome), List of Abbreviations, References.